# Nonthermal magnetization pathways in photoexcited semiconductors

Giovanni Marini[1,2*]

**1** Dipartimento di Fisica, Universitá di Trento, via Sommarive 14, I-38123 Povo, Italy
**2** Graphene Labs, Fondazione Istituto Italiano di Tecnologia, Via Morego, I-16163 Genova, Italy
* giovanni.marini-2@unitn.it

September 26, 2025

## Abstract

The stabilization of long-range magnetic order in nominally non-magnetic semiconductors using femtosecond light pulses is an exciting yet experimentally challenging goal. Theoretical studies indicate that certain non-magnetic semiconductors can exhibit transient magnetic instabilities following above-gap laser excitation, but the dynamical pathways leading to these states remain largely unexplored. In this work, I introduce a minimal real-time spin-orbital model and identify the fundamental microscopic mechanisms that enable the emergence of a transient magnetic order. I then discuss the relevance of these findings for real materials employing a phenomenological time-dependent Ginzburg-Landau model. Finally, I analyze the strengths and limitations of current first-principles methodologies for investigating dynamically induced broken-symmetry states in the light of the present results.

# 1  Introduction

Femtosecond laser pulses have the potential to enable the ultrafast control of magnetism and the design of magnetic devices with unprecedented operational speed and efficiency [1–3]. Ultrafast demagnetization [4] was the first evidence that all-optical control of the magnetic state at the femtosecond timescale is possible and opened to the observation of many related phenomena including all-optical switching [1, 2, 5–8], ultrafast magnetization reversal, light-induced magnetic phase transitions [9,10], light-enhanced magnetism [11–13] and all-optical control of ferromagnetism [14]. From the theoretical standpoint, ultrafast magnetic dynamics represents an extremely complex phenomenon. The fundamental role of spin-orbit coupling (SOC) in ultrafast demagnetization was soon recognized [15], however after almost 30 years no consensus exists on the role of concomitant microscopic mechanisms, including spin-orbit mediated transfer of angular momentum to the orbital part [16] (eventually lost via other scattering mechanisms e.g. phonons [17]), superdiffusive spin transport [18,19] and spin-mediated electron-phonon coupling [20,21]. A related yet more elusive phenomenon is the light-induced magnetization of non-magnetic materials, which has been observed experimentally employing circularly polarized light through the inverse Faraday effect [22–24], although the resulting magnetism is short lived and the non-magnetic nature of the irradiated material is not altered. More recently, an alternative dynamical pathway for the magnetization of non-magnetic materials based on the nuclear motion has also been demonstrated [25]. Some mechanisms to induce magnetism in non-magnetic materials have also been proposed theoretically: a mechanism to induce a robust transient ultrafast magnetism in non-magnetic semiconductors based on the spontaneous breaking of time-reversal symmetry after above-gap photoexcitation in the resulting quasi-equilibrium state was proposed in Ref. [26]. The resulting magnetic state is expected to live until carrier recombination. Another mechanism has been proposed to induce a ferromagnetic state in monolayers $MoSe_2$ interfaced with ferromagnetic $MnSe_2$ employing light, exploiting the spin transfer from the ferromagnetic layer [27]. Recent theoretical studies further showed that magnetic dynamics can also be initiated after off-resonant pumping due to time-local explicit time-reversal breaking [28] even with linearly-polarized light, or due to an explicitly time symmetry breaking carrier envelope [29]. All these investigations are supported by first-principles real-time time-dependent density functional theory (TDDFT) analyses and there is consensus that spin-orbit coupling plays a pivotal role for the initiation of spin dynamics. However, since most TDDFT implementations lack mechanisms for the relaxation of the photoexcited state [28], it is unclear to what extent can the TDDFT reproduce the magnetization dynamics of real materials, where instead the photoexcited relaxes through many mechanisms, including many-body electron-electron interaction and coupling to other degrees of freedom (phonons, photons). To shed light on the relaxation pathways of a photoexcited semiconductor hosting a transient magnetic instability, here I discuss a minimal spin-orbital Hamiltonian which includes the effect of the laser electric field, spin-orbit coupling, interactions and an energy relaxation mechanism. Importantly, I find that the non-dissipative dynamics (analogous to the TDDFT one) well describes the initial phase of the dynamics after laser irradiation, but a dissipative term is fundamental to capture the transition towards transient broken symmetry phases. Finally, I qualitatively analyze the expected trajectory of the magnetic order parameter in resonantly photoexcited semiconductors hosting a transient magnetic instability within a time-dependent Ginzburg-Landau model [30].

## 2 Modeling ultrafast spin dynamics

### 2.1 The spin-orbital model

The specific physical scenario that I aim to describe is a non-magnetic semiconductor illuminated with above-gap laser light. After above-gap photoexcitation, carriers populate the conduction band and a quasi-equilibrium electronic state is reached after electron thermalization but before electron recombination. It was shown that this quasi-equilibrium state may show a magnetic instability (for example, a ferrimagnetic instability in $V_2O_5$ [26]). The reason is that the electrons are "constrained" in the conduction manifold for a certain time before recombination [31], and thus the lowest energy state in the presence of such constraint may be different from the equilibrium one, giving rise to a multitude of electronic and structural transitions towards transient broken symmetry phases [26, 32–35, 35–38]. In the case of a magnetic instability in the photoexcited semiconductor, I show here that many essential features of the spin dynamics can be discussed within a toy model made of localized interacting spins, in the spirit of a small cluster Heisenberg model, and coupling it to one orbital angular momentum. More specifically, I propose a minimal spin-orbital model designed to describe the relaxation towards a transient magnetic photoexcited state, capturing the interplay between spin, orbital angular momentum, spin-spin interactions and spin-orbit coupling (SOC) under the influence of a time-dependent external perturbation. In this situation sever In order to model the system's state after photoexcitation, the spin-orbital system is initialized in an excited eigenstate. From an intuitive standpoint, this amounts to say that the insulating ground state of the unconstrained Hamiltonian may be regarded as an excited configuration of the constrained Hamiltonian. The model Hamiltonian introduced here can be represented in an Hilbert space consisting of four $s = 1/2$ spins, one of which is coupled to an $l = 1$ orbital angular momentum through spin-orbit coupling. The full Hilbert space dimension is thus $6 \times 2^3 = 48$. The Hamiltonian is written:

$$H(t) = H_{\text{SOC}} + H_{\text{int}} + H_{\text{kick}}(t), \tag{1}$$

where the spin-orbit coupling $H_{SOC}$ acts on the spin-orbital as:

$$H_{\text{SOC}} = \lambda \left( L_{1x} S_{1x} + L_{1y} S_{1y} + L_{1z} S_{1z} \right) \tag{2}$$

and $L_{1\alpha}$ and $S_{1\alpha}$ are orbital and spin operators for orbital and spin $l$, respectively, while $\lambda$ is the SOC strength. Spin-spin interactions couple the four spins with anisotropic exchange couplings $J_\alpha^{ij}$:

$$H_{\text{int}} = \sum_{i=1}^{4} \sum_{\alpha=x,y,z} J_\alpha^{ij} S_{i\alpha} S_{j\alpha} \tag{3}$$

The effect of the external time-dependent laser's electric field (I neglect the small laser magnetic field component) is modeled as a pulsed perturbation acting on all the orbital components:

$$H_{\text{kick}}(t) = f(t) \sum_{i=1}^{2} (a_x L_{ix} + a_y L_{iy} + a_z L_{iz}) \tag{4}$$

where the envelope $f(t)$ is taken as a half-cosine pulse:

$$f(t) = \begin{cases} A \cos\left( \dfrac{\pi(t - t_0)}{\tau} \right) & t_0 \leq t \leq t_0 + \tau \\ 0 & \text{otherwise} \end{cases} \tag{5}$$

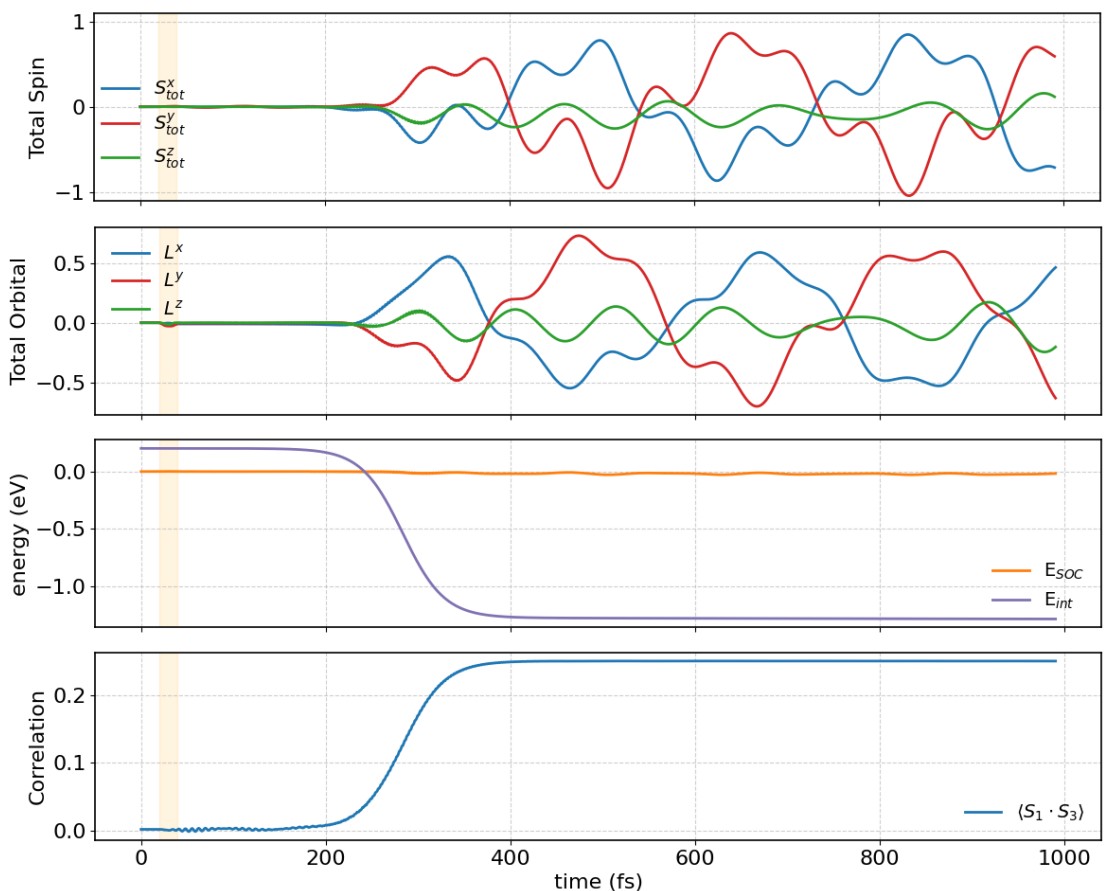

Figure 1: Real-time dynamics for the spin-orbital model in the dissipative case. From top to bottom: total spin components as a function of time, total orbital angular momentum components as a function of time, system energy $E_{int}$ as a function of time and the spin-orbit energy $E_{SOC}$, and correlation between $S_1$ and $S_3$, $\langle S_1 \cdot S_3 \rangle$. The yellow rectangles identify the time window when the laser perturbation $H_{kick}$ is active.

with amplitude $A$, onset $t_0$, and duration $\tau$, such that the product $f(t)\vec{L}$ is explicitly time-reversal symmetric. Details in the numerical values of the parameters are given in the methods. Physically, this mimics the effect of a linearly polarized laser, which has been shown to be able to transfer angular momentum also in the absence of explicit time-reversal symmetry breaking if the system's symmetry allows for it [26, 28, 29]. To model the energy loss of the excited quantum state without having to couple it to additional degrees of freedom (phonons, photons, defects), I include a phenomenological quantum friction term in the time evolution similarly to the approach presented in Ref. [39]. In this approach the state vector evolves according to a non-unitary Schrödinger-type equation that breaks time-reversal invariance and cools the system:

$$i\frac{\partial}{\partial t}\left|\psi(t)\right\rangle = H(t)\left|\psi(t)\right\rangle - i\,\eta\left[H(t) - \langle H(t)\rangle\right]\left|\psi(t)\right\rangle \tag{6}$$

where $\langle H(t)\rangle = \langle\psi(t)|H(t)|\psi(t)\rangle$ is the instantaneous energy expectation value, and $\eta > 0$ is a small dimensionless damping parameter, fixed to $\eta = 0.01$ when not differently specified. Eq. (6) is equivalent to time evolution under an effective non-Hermitian generator:

$$H_{\text{eff}}(t) \;=\; H(t) - i\,\eta\left[H(t) - \langle H(t)\rangle\right] \tag{7}$$

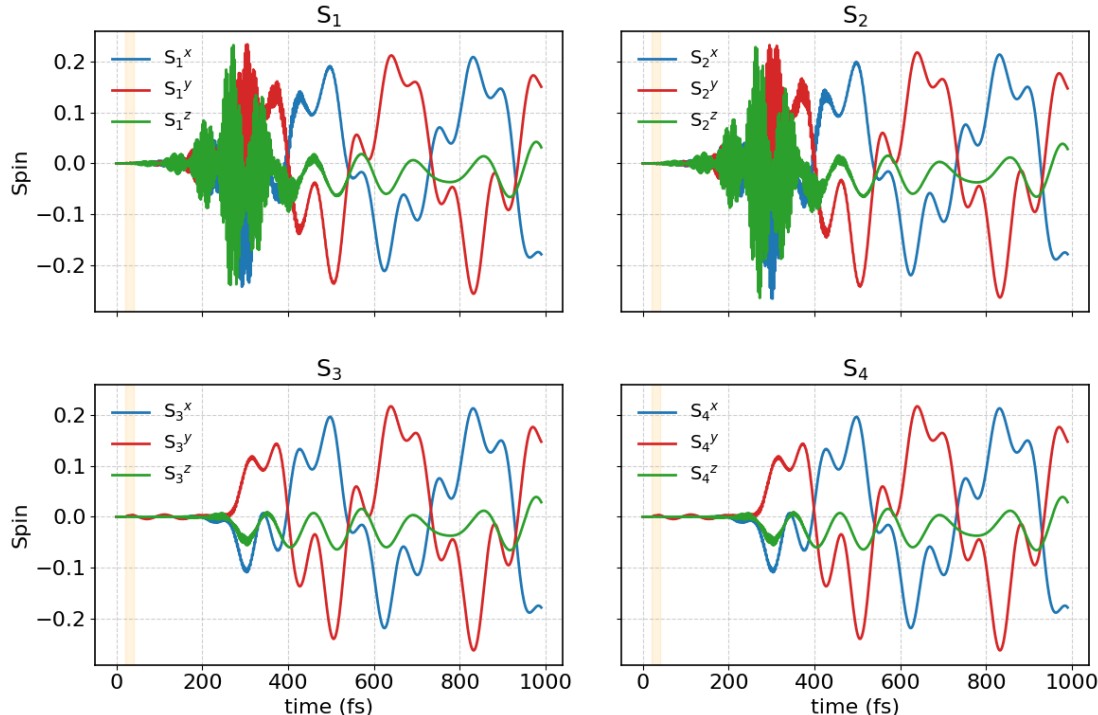

Figure 2: Spin dynamics in the dissipative case. In each of the four panels the spin components of the corresponding spin are reported. The yellow rectangles identify the time window when the laser perturbation $H_{kick}$ is active.

Because the non-Hermitian part is proportional to the Hamiltonian itself the dynamics preferentially damps components of the wavefunction with energy above the instantaneous mean energy: namely, in the instantaneous eigenbasis $H|\phi_n\rangle = E_n|\phi_n\rangle$ the amplitudes $c_n(t)$ satisfy to leading order

$$\dot{c}_n(t) = -\eta\big(E_n - \langle H(t)\rangle\big)\,c_n(t) \quad \Rightarrow \quad c_n(t) \propto \exp\!\Big[-\eta\int^t \big(E_n - \langle H(t')\rangle\big)\,dt'\Big] \qquad (8)$$

so that high-energy components decay faster than low-energy ones and the state is driven toward the lowest-energy sector compatible with symmetries and the initial conditions. The quantum-friction form is phenomenological but it reflects what it is expected to happen in open-system dynamics after visible photoexcitation, where electrons are expected to lose part of the absorbed energy through electron-phonon scattering. Additional technical details on the model are given in the methodological Appendix A.

## 2.2 Laser-induced spin-orbital dynamics

In Fig.1 the main observables characterizing the spin-orbital dynamics are depicted. After the laser acts (yellow rectangle in the figure) and triggers a small change to the orbital angular momentum (order of 0.01 in units of $\hbar$), the system remains very close to its initial energy for approximately 150 fs. Nevertheless, a spin dynamics is initiated by the laser in combination with the action of SOC on spin $S_1$ (the only spin coupled to orbital angular momentum): this can be observed in Fig.2, where a rather chaotic spin dynamics is initially shown. After approximately 250 fs the total spin suddenly starts showing strong oscillations and $\approx 1000$ fs after the pulse, the system has relaxed down to $-1.28$ eV, in

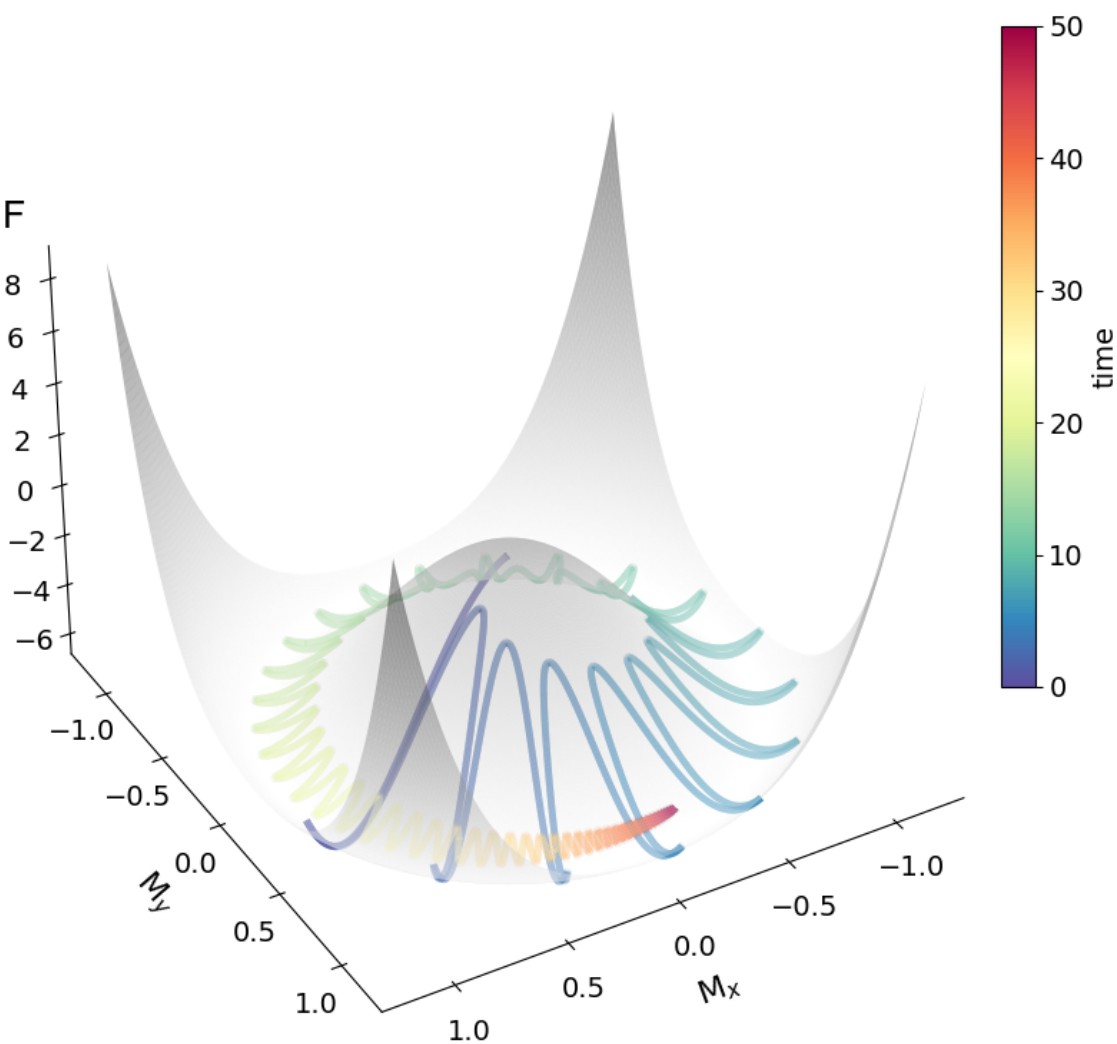

Figure 3: Free energy surface (gray shades) and real-time magnetization trajectory for the time-dependent Ginzburg-Landau model. Hotter colors indicate later times in the trajectory.

close proximity to the ground state, whose energy is calculated to be approximately $-1.29$ eV. The spins start from an almost uncorrelated configuration, as shown in the lower panel of Fig.1. As the energy starts decreasing, the correlation between the four spins becomes almost perfect. This is monitored by plotting the correlator between two spins, $S_1$ and $S_3$, $\langle S_1 \cdot S_3 \rangle$. The correlated dyanmics between of the four spins at late times is also evident in Fig.2 and signals the transition to a different quantum state. Despite playing a fundamental role in initiating the spin dynamics, the SOC contribution to total energy remains very small for the whole dynamics (third panel in Fig.1): this is consistent with previous observations in ultrafast demagnetization [40] and is ultimately regulated by the value of the $\lambda$ parameter. I note that this model does not conserve total angular momentum during the dynamics due to the combination of anisotropic exchange couplings (see Appendix A). From the tests performed by suppressing the laser amplitude $A$, the spin-orbit coupling $\lambda$ and the dissipation $\eta$, I find that the spin dynamics emerges from the combined effect of the three. I thus suggest that as long as the laser can transfer angular momentum to the spin system (see also the discussions of Refs. [26, 28, 29]) one should expect the onset of a spin dynamics also in the more complex Hamiltonian describing real

materials and that, if a low-energy constrained broken symmetry state exists, it can be reached. This conclusion does not hold in the dissipationless case ($\eta = 0$), where I find that the spin dynamics is still activated by the laser but the spin-orbital system remains close to its initial state (see Appendix B for a detailed analysis). Importantly, this suggests that real-time TDDFT should only be able to access photoinduced broken symmetry low-energy states if some dissipation mechanism is added.

## 2.3  Time-dependent Ginzburg-Landau model for the order parameter

In the light of the results from the spin-orbital model I present in Fig.3 a possible model for the order parameter dynamics in semiconductors developing a magnetic instability after resonant photoexcitation, based on a two-dimensional isotropic time-dependent Ginzburg-Landau model [30]. While the precise dynamics will depend on experiment- and material-specific properties, some general purely qualitative features are still worth discussing. After above-gap photoexcitation I assume that the order parameter dynamics is either initiated by a thermodynamical fluctuation of the order parameter $\mathbf{M}$ or by fluctuations in the spin expectation value resulting from orbital angular momentum injection, via the mechanism just discussed in the the spin-orbital model. In the light of previous results, it is reasonable to assume that a transverse order parameter oscillation like the one observed in the spin-orbital model should also be included, causing a rotating motion of the order parameter in time. A phenomenological damping $\gamma$ is also included (see Appendix A for all the numerical details). The order parameter trajectory naturally ends up in the isotropic minimum of the Ginzburg-Landau potential. Since the model is isotropic, the final orientation of $\mathbf{M}$ is only determined by the initial condition. I note that in a more realistic scenario, many other factors will contribute to the final orientation of $\mathbf{M}$ (free-energy anisotropy, magnetic impurities, structural defects). If the initial fluctuation is due to a thermodynamical fluctuation, it should be randomly directed in the case of an isotropic system. In this case, one should expect the formation of domains with different magnetization orientation and possible topological defects at the domain walls, according to the Kibble-Zurek mechanism [41,42]. A similar behavior has already been hypothesized to explain amplitude-mode damping in a photoinduced charge-density wave transition measured experimentally [43]. Conversely, if the initial fluctuation of the spin is caused by the orbital angular momentum injection from the laser, the final magnetization orientation could in principle be coherent across the illuminated sample.

# 3  Conclusion

In conclusion I studied the real-time dynamics of a spin-orbital model and analyzed the role of various Hamiltonian terms relevant for the insurgence of a magnetic dyanmics in a non-magnetic system. I found that a combination of spin-orbit coupling and orbital angular momentum injection can trigger a sizable spin dynamics, in agreement with previous TDDFT results in real materials. Importantly, I found that in the absence of dissipation the model spin-orbital system remains in the excited state indefinitely, suggesting that magnetic phase transitions are likely to be inaccessible in TDDFT at finite simulation time without adding some dissipative mechanism. I speculate that a spin dynamics similar to the one described here will also be present in the more complicated Hamiltonian of a photoexcited semiconductor hosting a transient magnetic instability, influencing the timescale to reach the constrained quasi-equilibrium low energy state. Finally, I presented a phenomenological time-dependent Ginzburg-Landau model and analyzed a possible magnetization trajectory in the presence of tangential spin oscillations, which represents a

useful paradigm to interpret future ultrafast magnetization experiments.

# Acknowledgements

I am grateful to Dr. Matteo Furci, Dr. Stefano Mocatti, Dr. Andrea Corradini, Prof. Pierluigi Cudazzo and Prof. Matteo Calandra for insightful scientific discussions.

**Funding information** Funded by the European Union (ERC, DELIGHT, 101052708). Views and opinions expressed are however those of the authors only and do not necessarily reflect those of the European Union or the European Research Council. Neither the European Union nor the granting authority can be held responsible for them.

# A  Methods

## A.1  Spin-orbital model

The Hilbert space for the Hamiltonian $H(t)$

$$H(t) = H_{\text{SOC}} + H_{\text{int}} + H_{\text{kick}}(t), \tag{9}$$

is constructed in a direct product basis:

$$\left[ (|m_l, m_s\rangle_1 \otimes (|\uparrow\rangle, |\downarrow\rangle)_2 \otimes (|\uparrow\rangle, |\downarrow\rangle)_3 \otimes (|\uparrow\rangle, |\downarrow\rangle)_4 \right], \tag{10}$$

where $m_l = \{-1, 0, +1\}$ and $m_s = \{\uparrow, \downarrow\}$. This yields a 48-dimensional basis. Operators are embedded into the full Hilbert space via Kronecker products as

$$L_{1z} = L_z \otimes \mathbb{I}_2 \otimes \mathbb{I}_2 \otimes \mathbb{I}_2 \otimes \mathbb{I}_2, \quad S_{3x} = \mathbb{I}_6 \otimes \mathbb{I}_2 \otimes \sigma_x \otimes \mathbb{I}_2, \tag{11}$$

and so on, where $\mathbb{I}_n$ is the $n$-dimensional identity operator. For the interaction term $H_{int}$

$$H_{\text{int}} = \sum_{i=1}^{4} \sum_{\alpha=x,y,z} J_\alpha^{ij} S_{i\alpha} S_{j\alpha} \tag{12}$$

The spin-orbit coupling parameter is fixed to $\lambda = 0.05$ eV. The following values for the exchange couplings are employed:

$$\begin{aligned}
J_x^{13} = J_x^{14} = J_y^{13} = J_y^{14} = J_z^{13} = J_z^{14} &= -1 \text{ eV} \\
J_x^{23} = J_x^{24} = J_y^{23} = J_y^{24} = J_z^{23} = J_z^{24} &= -1 \text{ eV} \\
J_x^{12} = J_x^{34} = J_y^{12} = J_y^{34} &= -0.5 \text{ eV} \\
J_z^{12} = J_z^{34} &= -0.4 \text{ eV}
\end{aligned} \tag{13}$$

having added a small $z$-anisotropy in the $1-2$ and $3-4$ interaction channels to prevent total momentum conservation when adding the dissipative term, similarly to what one would expect in a more realistic scenario (angular momentum exchange with phonons) and to what one usually observes in first-principles real-time simulations, where total angular momentum is never explicitly conserved [44]. For the kick Hamiltonian I choose $a_x = a_y = a_z = 1$ and $t_0 = 30$ fs, $\tau = 30$ fs and $A = 0.1$ eV$^{-1}$ (in $\hbar$ units) for the envelope function in the kick term. The time-dependent Schrödinger equation is solved as

$$i\hbar \frac{\partial}{\partial t} |\psi(t)\rangle = H(t)|\psi(t)\rangle, \tag{14}$$

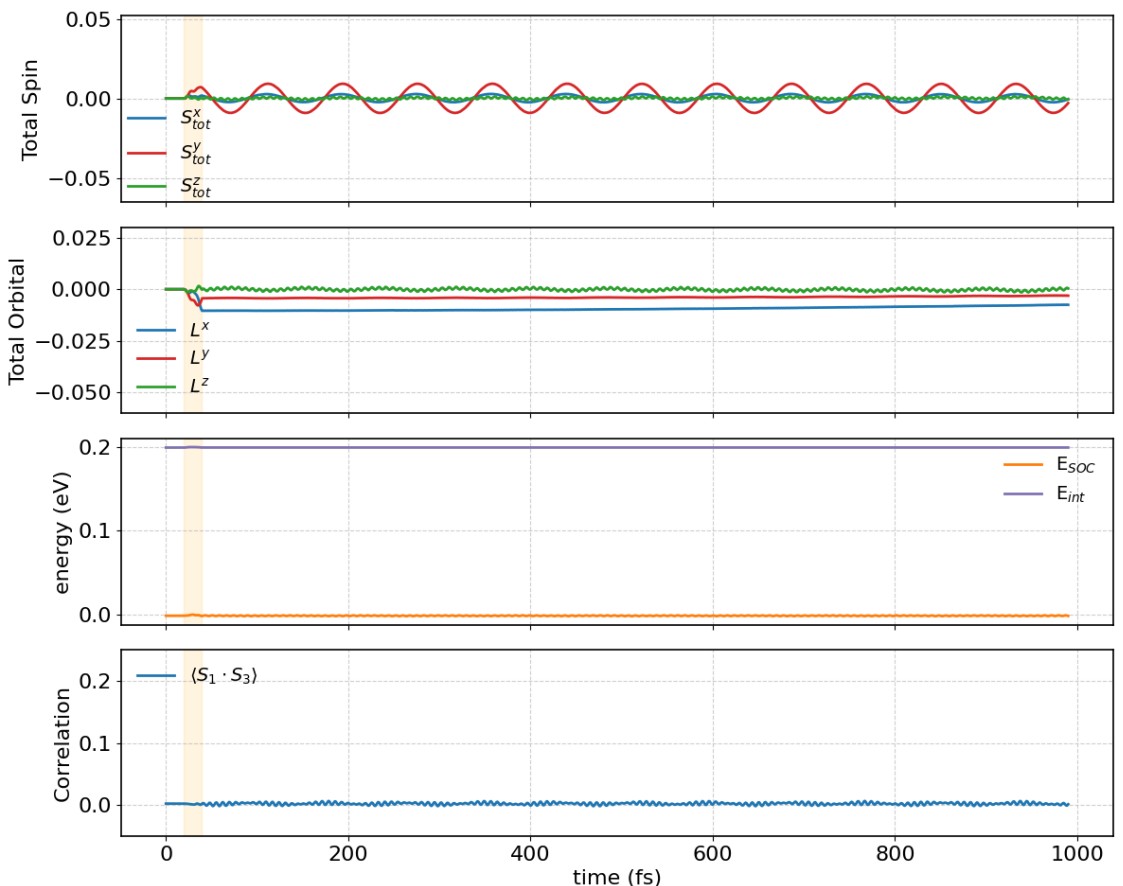

Figure 4: Real-time dynamics for the spin-orbital model in the dissipationless case. From top to bottom: total spin components as a function of time, total orbital angular momentum components as a function of time, system energy $E_{int}$ as a function of time and the spin-orbit energy $E_{SOC}$, and correlation between $S_1$ and $S_3$, $\langle S_1 \cdot S_3 \rangle$. The yellow rectangles identify the time window when the laser perturbation $H_{kick}$ is active.

with $\hbar$ set to unity. I assume the Hamiltonian to be expressed in eV, meaning that the natural unit of time is $\hbar/\text{eV} \approx 0.66$ fs. Numerical integration is performed using the `solve_ivp` routine from `SciPy` [45], employing an adaptive Runge-Kutta scheme with absolute tolerance $10^{-9}$ and relative tolerance $10^{-7}$. The initial state is chosen to be an excited state of the Hamiltonian $E_n$, such that $E_n - E_0$ is of the order of the eV. Expectation values of spin and orbital operators are evaluated as

$$\langle O \rangle(t) = \langle \psi(t)|O|\psi(t)\rangle, \tag{15}$$

Evolution under Eq. (6) is non-unitary and the norm $\langle \psi|\psi \rangle$ decays in time. To maintain the norm the $|\psi(t)\rangle$ is renormalized after each time step. The quantum-friction operator is proportional to $H$ and therefore preserves any symmetries of $H$. The time evolution does not however generally preserve expectation values of quantities not commuting with the Hamiltonian, including total angular momentum $J_{tot}$ in the case of anisotropic coupling in $H_{int}$.

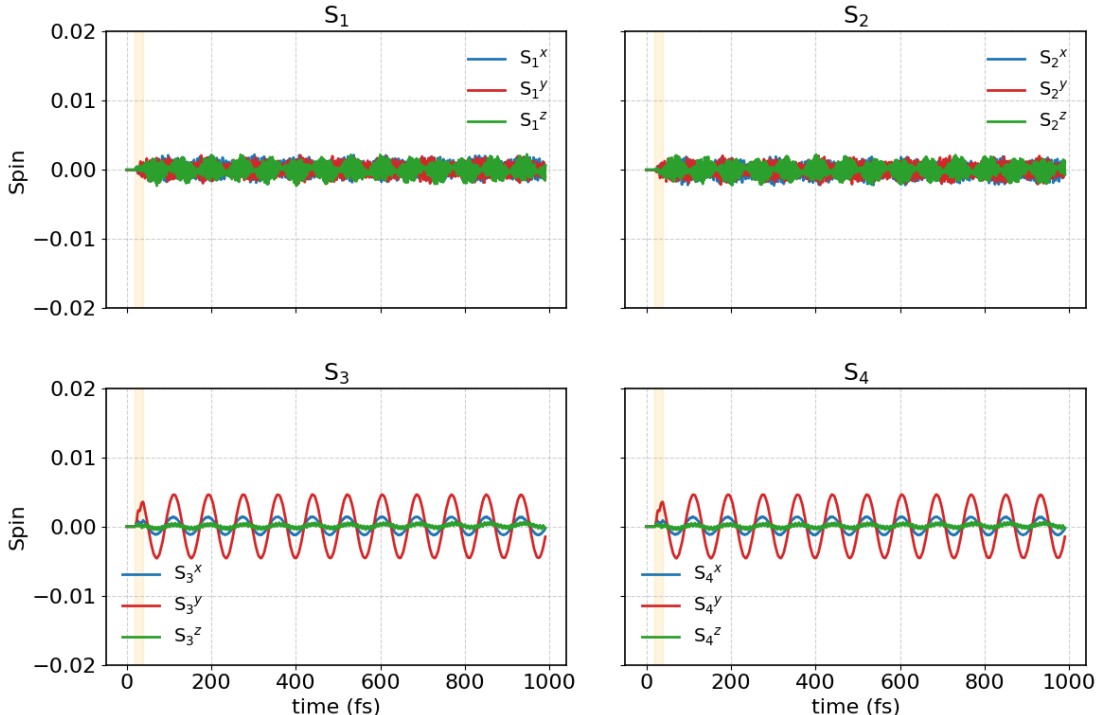

Figure 5: Spin dynamics in the dissipationless case. In each of the four panels the spin components of the corresponding spin are reported. The yellow rectangles identify the time window when the laser perturbation $H_{kick}$ is active.

## A.2    Time-dependent Ginzburg-Landau model

To study the ultrafast emergence of a magnetic state from a nonmagnetic configuration, I model the dynamics of the magnetic order parameter $\mathbf{M}(t)$ using a time-dependent Ginzburg-Landau model. Near the symmetric state, the free energy $F(\mathbf{M})$ can be approximated by a quadratic expansion, such that a negative curvature at $|\mathbf{M}| = 0$ defines the instability:

$$\Gamma^2 = \left.\frac{d^2 F}{d|\mathbf{M}|^2}\right|_{|\mathbf{M}|=0} < 0, \tag{16}$$

I model the ultrafast dynamics of the magnetic order parameter in terms of its two in-plane components $\mathbf{M}(t) = (M_x(t), M_y(t))$. The effective free-energy functional is taken to be of Landau form

$$F[\mathbf{M}] = \tfrac{1}{2}\Gamma^2|\mathbf{M}|^2 + \tfrac{\alpha}{4}|\mathbf{M}|^4, \tag{17}$$

which has a symmetric Mexican-hat shape below the critical point, with $\alpha > 0$ representing the positive coefficient in front of the quartic term. I consider an equation of motion for $\mathbf{M}(t)$ is of the form:

$$\ddot{\mathbf{M}} = \Gamma^2\mathbf{M} + \alpha|\mathbf{M}|^2\mathbf{M} - \gamma\dot{\mathbf{M}} + \mathbf{a}_o(t), \tag{18}$$

a phenomenological damping constant $\gamma$ and an orthogonal second derivative component $\mathbf{a}_o(t)$ to mimic the spin rotation have been added. The time propagation is performed numerically by discretizing time in steps $\Delta t$ and updating both $\mathbf{M}(t)$ and its velocity $\dot{\mathbf{M}}(t)$ iteratively. In practice, I employ a second-order Velocity-Verlet-like finite-difference

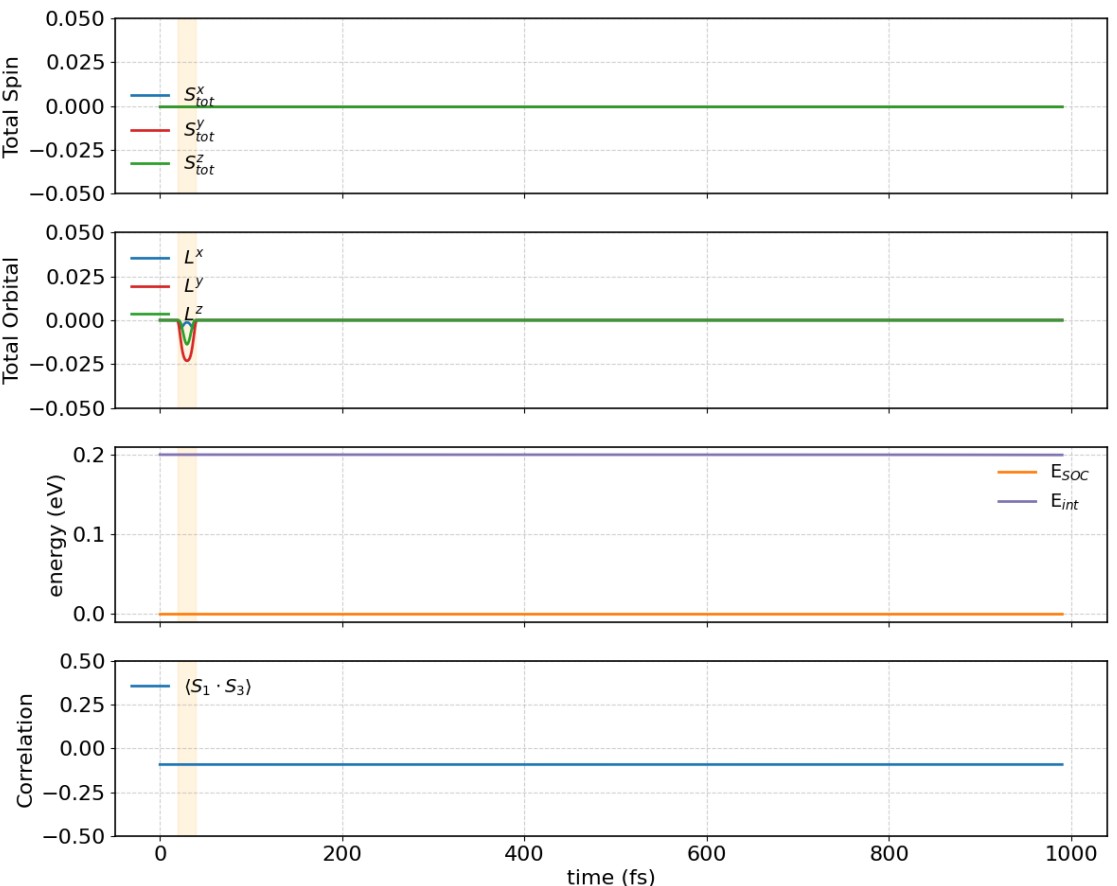

Figure 6: Real-time dynamics for the spin-orbital model with $\lambda = 0$.

scheme. The update rules are

$$\dot{\mathbf{M}}(t + \Delta t) = \dot{\mathbf{M}}(t) + \ddot{\mathbf{M}}(t)\,\Delta t, \tag{19}$$

$$\mathbf{M}(t + \Delta t) = \mathbf{M}(t) + \dot{\mathbf{M}}(t + \Delta t)\,\Delta t. \tag{20}$$

This approach allows to track the trajectory of the order parameter from the unstable symmetric point at $\mathbf{M} = 0$ toward the ring of minima, while capturing both radial relaxation and tangential motion, as well as their dissipation. For the simulation in Fig.3 I used the values (arbitrary units):

$$\begin{aligned}\Gamma = 5, \quad \alpha = 30, \quad \gamma = 0.15, \quad t_{max} = 50 \\ \mathbf{a}_o(t) = e^{-0.1t}\,\mathbf{a}_o(0), \qquad \mathbf{a}_o(0) = 0.1\end{aligned} \tag{21}$$

while the initial condition is:

$$\begin{cases} M_x(0) = 0.01 \qquad M_y(0) = 0 \\ \dot{M}_x(0) = 0 \qquad \dot{M}_y(0) = 0 \end{cases} \tag{22}$$

## B    The role of $\eta$, $\lambda$ and $A$ in the spin-orbit model

In Fig.4 and Fig.5 I report the spin-orbital real-time dynamics in the absence of dissipation ($\eta = 0$), again initializing the system in the same excited eigenvector as in the dynamics

shown in Fig.1. Also in this case, the laser-induced angular momentum initiates a sizable spin dynamics of the same order of magnitude of the one observed in the dissipative case during the first 200 fs. However, in this case the system remains locked in an excited state at constant energy and cannot reach the ground state at longer times, as demonstrated by the behavior of the $\langle S_1 \cdot S_3 \rangle$ correlator. A qualitatively similar behavior should be expected in first-principles TDDFT simulations without dissipation, compatibly with previously reported results [26, 28, 29]. Finally, in Fig.6 I report the time-evolution for the spin-orbital model in the dissipative case in the absence of SOC. Without SOC the eigenvectors are different; thus I choose a linear combination of eigenvectors for the new Hamiltonian having comparable energy as the one employed for the other simulations. No meaningful dynamics is observed for the first 1000 fs, demonstrating the fundamental role played by the SOC term in magnetic dynamics. I also verified that, similarly, no spin dynamics is activated when $A = 0$, i.e. no laser action (not shown).

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
