# Peer review of "Nonthermal magnetization pathways in photoexcited semiconductors"

_SciPost Physics_

## Round 1 · Referee Report · Anonymous (Referee 1) · 2025-11-21

Report

The MS by Marini theoretically explores light-induced magnetic states in initially non-magnetic materials with a combined model Hamiltonian approach, and a GL formulation. The main novel results are in the insight showing that a kicked angular momentum completely suffices to initiate the process of magnetization, and analyzing the contributions of the different interactions to the dynamics. Moreover, the role of relaxation is incorporated and shown to lead to slow dephasing of the magnetic state, which is usually not modeled in TDDFT and ab-initio simulations of ultrafast magnetization.
Overall, I think that the insight obtained in the simple model is extremely interesting, and definitely worthy of publication in scipost. In particular, it might end up motivating novel schemes for optically controlling magnetism, or elucidating the physical mechanism responsible in different systems. Therefore, I strongly support publication after a minor revision addressing the comments below:
1. I think that some of the key results are not sufficiently highlighted in the abstract and introduction currently. The intro is quite long and the context of the main results here is not clearly stated there. Therefore, I think it could be great if the author could emphasize these points. Especially, the novelty in the idea to only kick the electronic angular momentum to simulate the process.
2. It would be great to move some of the technical details, especially the main equations of motion, to the main text. Unless there’s a specific requirement by scipost, it would help to see the equations for the model while analyzing the figures.
3. What is the importance of the type of kick given to L – in terms of intensity, direction, polarization axis?
4. I think some discussion on the limitations of the model used here is still needed – for instance, in TDDFT one includes directly the interaction between light and electrons, while here that is completely neglected and replaced by an angular momentum kick. That captures the main details nicely, but might fail in certain regimes, e.g. if e-e interactions cause a dissipation of the electronic angular momentum, or many other features neglected. Those should be highlighted. Similarly, spin-phonon coupling that could be responsible for several mechanisms of light induced magnetism is not included.
5. It should be explained what kind of relaxation pathways are included here – there are many options, and which are not.
6. Fig. 3 currently doesn’t contribute a whole lot to the text. It might be better to find a more quantitative way to plot the data from the GL model, especially, to compare it directly to the model spin simulations.
7. The long timescale periodic spin dynamics (e.g. in fig 4) seem reminiscent of some magnonic excitation, perhaps also a superposition of magnons. Is that the case? It’s worth discussing.
8. The main idea presented by this paper as I see it is that in order to induce strong control over magnetization, one needs to find a material where laser-L coupling is strong – it’s worth discussing what good options might be.

Recommendation

Publish (easily meets expectations and criteria for this Journal; among top 50%)

  • validity: high
  • significance: good
  • originality: high
  • clarity: high
  • formatting: perfect
  • grammar: perfect

Author:  Giovanni Marini  on 2025-12-22  [id 6170]

(in reply to Report 1 on 2025-11-21)

I sincerely thank the Referee for having assessed my manuscript. Please find my response to all points below.

Referee: "The MS by Marini theoretically explores light-induced magnetic states in initially non-magnetic materials with a combined model Hamiltonian approach, and a GL formulation. The main novel results are in the insight showing that a kicked angular momentum completely suffices to initiate the process of magnetization, and analyzing the contributions of the different interactions to the dynamics. Moreover, the role of relaxation is incorporated and shown to lead to slow dephasing of the magnetic state, which is usually not modeled in TDDFT and ab-initio simulations of ultrafast magnetization.
Overall, I think that the insight obtained in the simple model is extremely interesting, and definitely worthy of publication in scipost. In particular, it might end up motivating novel schemes for optically controlling magnetism, or elucidating the physical mechanism responsible in different systems. Therefore, I strongly support publication after a minor revision addressing the comments below:"
Author: I sincerely thank the Referee for the careful reading of my manuscript and for the positive assessment of my work. I are grateful for the suggestions, which have helped me improve the clarity and presentation of the results. Below I address all comments point by point. In the revised manuscript, all corresponding changes have been implemented.

Referee: "1. I think that some of the key results are not sufficiently highlighted in the abstract and introduction currently. The intro is quite long and the context of the main results here is not clearly stated there. Therefore, I think it could be great if the author could emphasize these points. Especially, the novelty in the idea to only kick the electronic angular momentum to simulate the process."
Author: I thank the Referee for pointing this out. In the revised version, I have modified both the abstract and introduction to more clearly emphasize: (i) the conceptual novelty of initiating magnetization solely through an impulsive kick of the electronic orbital angular momentum and (ii) the importance of relaxation in governing the long-time decay of the induced magnetic state. These points are now stated explicitly and concisely. I also expanded the relevant abstract sentence, as indicated in the list of changes.

Referee: "It would be great to move some of the technical details, especially the main equations of motion, to the main text. Unless there’s a specific requirement by scipost, it would help to see the equations for the model while analyzing the figures."
Author: I think that the Referee's suggestion is on point. I moved the equations of motion from the Appendix directly into the main text also for the Ginzburg-Landau part, while keeping more technical details in the Appendix. This should improve readability.

Referee: "3. What is the importance of the type of kick given to L – in terms of intensity, direction, polarization axis?"
Author: I thank the Referee for this question. In short, the intensity determines the magnitude of the initial driving and consequently affects the relaxation time of the excited state. The pulse polarization axis determines the initial spin dynamics through H_SOC and is anticipated to play a central role in strongly anisotropic systems. I have expanded the discussion of how the kick amplitude, direction, and polarization affect the resulting magnetization dynamics at the end of Appendix B. I also added a new figure, Fig.7, where I explicitly study the role of pulse intensity.

Referee: "4. I think some discussion on the limitations of the model used here is still needed – for instance, in TDDFT one includes directly the interaction between light and electrons, while here that is completely neglected and replaced by an angular momentum kick. That captures the main details nicely, but might fail in certain regimes, e.g. if e-e interactions cause a dissipation of the electronic angular momentum, or many other features neglected. Those should be highlighted. Similarly, spin-phonon coupling that could be responsible for several mechanisms of light induced magnetism is not included."
Author: I agree with the Referee in that a more detailed discussion of the model's limitations is due. In order to answer both to this question and question No. 5, I added a new subsection to the manscript (now 2.3), named: "Significance and limitations of the present model", where I discuss both the limitation of this model when compared to a more realistic study which includes microscopic interactions.

Referee:"5. It should be explained what kind of relaxation pathways are included here – there are many options, and which are not."
Author: See the response to point 4.

Referee:" 6. Fig. 3 currently doesn’t contribute a whole lot to the text. It might be better to find a more quantitative way to plot the data from the GL model, especially, to compare it directly to the model spin simulations."
Author: I thank the Referee for commenting on this. My aim with Figure 3 is not to compare directly with the spin model, but rather to give a general idea of what could be happening in a real material. For this reason, I chose to present the data only qualitatively through the trajectory presented in Figure 3. However, from the Referee's comment I understood that my intention was not clear at all in the present form. For this reason, I modified the discussion in the Ginzburg-Landau model Sec. 2.4 (old Sec 2.3) so to clarify the illustrative the scope of Fig 3. I believe that this, together with the changes from point 2., should be sufficient to give a clearer presentation of the TD Ginzburg-Lnadau simulation.

Referee:"7. The long timescale periodic spin dynamics (e.g. in fig 4) seem reminiscent of some magnonic excitation, perhaps also a superposition of magnons. Is that the case? It’s worth discussing."
Author: The Referee's observation about the similarity between the long-time behavior of spin in the model and magnons is spot on. The long-time oscillations possessing well-defined frequency in Fig.4 and Fig.5 originate from the coherent evolution of the excited state. The physical origin is the following: the excited state is not an eigenstate of H_int in general, giving rise to a non-trivial spin dynamics. Such dynamics is collective and comes from the state evolution under "H_int+H_SOC", and leads to persistent oscillations at a discrete frequency for spin 3 and 4. This behavior presents analogies to coherent magnon oscillations in small magnetic clusters, where magnons are quantized spin-wave levels rather than propagating modes. In both cases, both the spin and the total energy of the spin system, "E_int^0 (exp. value of H_int) oscillate in time. This discussion was added to Appendix A, when commenting on Figures 4 and 5.

Referee:"8. The main idea presented by this paper as I see it is that in order to induce strong control over magnetization, one needs to find a material where laser-L coupling is strong – it’s worth discussing what good options might be."
Author: I thank the Referee for this insightful suggestion. Let me explicitly state here that the main idea of this work is twofold: first, to shed light on why a spin dynamics emerges at all in non-magnetic semiconductors, and second, to discuss the microscopic mechanisms underlying this dynamics. Desirable material properties to observe a non-trivial dynamics include symmetry-allowed angular momentum transfer to electrons, and the presence of a photoinduced magnetic instability, in order to observe a trajectory like the one from the TD Ginzburg-Landau model. Platforms hosting a combination of the two could be systematically identified through high-throughput search. Following the Referee's idea I expanded the conclusion to include this important perspective, which could stimulate further work on the topic.

---

## Round 1 · Referee Report · Anonymous (Referee 2) · 2025-12-19

Strengths

Detailed modelling.

Weaknesses

Writing and presentation.

Report

In this manuscript, the author provided a spin-orbital Hamiltonian to understand light-induced dynamics based on time-dependent Ginzburg Landau theory. The results are interesting. However, I have a few minor comments for the author to address before I could recommend its publication:

(1) In Section 2.1, there is an unfinished sentence: "In this situation sever In order to model the system’s state after photoexcitation".

(2) In Eq. (4), what is ax/ay/az? Why the orbital components get linear response? Why i = 1,2 instead of 1-4?

Requested changes

See the comments (1) and (2) in the report.

Recommendation

Ask for minor revision

  • validity: good
  • significance: good
  • originality: high
  • clarity: good
  • formatting: acceptable
  • grammar: reasonable

Author:  Giovanni Marini  on 2025-12-22  [id 6171]

(in reply to Report 2 on 2025-12-19)

Referee: "In this manuscript, the author provided a spin-orbital Hamiltonian to understand light-induced dynamics based on time-dependent Ginzburg Landau theory. The results are interesting. However, I have a few minor comments for the author to address before I could recommend its publication:"
Author: I thank the Referee for the positive evaluation of the work. Below I answer their comments:

Referee:"(1) In Section 2.1, there is an unfinished sentence: ”In this situation sever In order to model the system’s state after photoexcitation”.
Author: I thank the Referee for spotting that. The beginning of the phrase was mistakenly included from an older version of the manuscript. I have now removed it.

Referee:"(2) In Eq. (4), what is ax/ay/az? Why the orbital components get linear response? Why i = 1,2 instead of 1-4?"
Author:I thank the Referee for this question. Indeed, there is an error in Eq.4: the ”kick” Hamiltonian should not present any summation over angular momenta, as only the first constituent possesses an orbital component. The correct version of Eq. (4) is:

H_kick(t) = f (t) (a_x L1x + a_y L1y + a_z L1z )

I have included the correct version in the revised version of the manuscript. The reason why I only couple one of the four spins with the external field is that this is the minimal setup to obtain a spin dynamics. In a more realistic scenario, all spins would be coupled, yet the qualitative conclusions would not change. As for a_x,a_y,a_z , they are added to the model to have the possibility of an anisotropic coupling between the laser pulse and the orbital angular momentum. The microscopic mechanisms allowing for a finite orbital angular momentum linear response are discussed e.g. in Ref. https://journals.aps.org/prl/abstract/10.1103/75tm-3t9b among the others. The linearity of the response is assumed in the simple toy model presented here but non-linear order responses can be expected in a realistic scenario.

Anonymous on 2025-12-22  [id 6172]

(in reply to Giovanni Marini on 2025-12-22 [id 6171])

I recommend its publication as the author fully addressed my concerns.

---

## Editorial Decision

resubmitted